# Clinical and molecular characterization of patients fulfilling Chompret criteria for Li-Fraumeni syndrome in Southern Brazil

Camila Matzenbacher Bittar [1,2☉], Yasminne Marinho de Araújo Rocha[2☉], Igor Araujo Vieira[1,2☉], Clévia Rosset[1,2‡], Tiago Finger Andreis[1,2‡], Ivaine Tais Sauthier Sartor[3], Osvaldo Artigalás[3], Cristina B. O. Netto[4], Barbara Alemar[1,2], Gabriel S. Macedo[2], Patricia Ashton-Prolla[1,2,4] *

**1** Programa de Pós-Graduação em Genética e Biologia Molecular (PPGBM), Universidade Federal do Rio Grande do Sul (UFRGS), Porto Alegre, Rio Grande do Sul, Brazil, **2** Laboratório de Medicina Genômica, Centro de Pesquisa Experimental, Hospital de Clínicas de Porto Alegre (HCPA), Porto Alegre, Rio Grande do Sul, Brazil, **3** Hospital Moinhos de Vento (HMV), Porto Alegre, Rio Grande do Sul, Brazil, **4** Serviço de Genética Médica, Hospital de Clínicas de Porto Alegre (HCPA), Porto Alegre, Rio Grande do Sul, Brazil

☉ These authors contributed equally to this work.
‡ These authors also contributed equally to this work.
* pprolla@gmail.com

## Abstract

Li-Fraumeni syndrome (LFS) is an autosomal dominant cancer predisposition syndrome caused by pathogenic germline variants in the *TP53* gene, characterized by a predisposition to the development of a broad spectrum of tumors at an early age. The core tumors related to LFS are bone and soft tissue sarcomas, premenopausal breast cancer, brain tumors, adrenocortical carcinomas (ACC), and leukemias. The revised Chompret criteria has been widely used to establish clinical suspicion and support *TP53* germline variant testing and LFS diagnosis. Information on *TP53* germline pathogenic variant (PV) prevalence when using Chompret criteria in South America and especially in Brazil is scarce. Therefore, the aim of this study was to characterize patients that fulfilled these specific criteria in southern Brazil, a region known for its high population frequency of a founder *TP53* variant c.1010G>A (p. Arg337His), as known as R337H. *TP53* germline testing of 191 cancer-affected and independent probands with LFS phenotype identified a heterozygous pathogenic/likely pathogenic variant in 26 (13.6%) probands, both in the DNA binding domain (group A) and in the oligomerization domain (group B) of the gene. Of the 26 carriers, 18 (69.23%) were R337H heterozygotes. Median age at diagnosis of the first tumor in groups A and B differed significantly in this cohort: 22 and 2 years, respectively (*P* = 0.009). The present study shows the clinical heterogeneity of LFS, highlights particularities of the R337H variant and underscores the need for larger collaborative studies to better define LFS prevalence, clinical spectrum and penetrance of different germline *TP53* pathogenic variants.

## Introduction

Li-Fraumeni (LFS) syndrome is an autosomal dominant cancer predisposition disorder mainly caused by pathogenic and likely pathogenic germline variants (PV) in the *TP53* tumor

**Data Availability Statement:** All relevant data are within the manuscript and its S1 Fig and S1,S2 Tables files.

**Funding:** This study was funded by grants from Conselho Nacional de Desenvolvimento Científico e Tecnológico (CNPq) (Grant # 478430/2012-4), Fundação de Amparo à Pesquisa do Estado do Rio Grande do Sul (FAPERGS) (Grant # 16/2551-0000486-2), and Fundo de Incentivo à Pesquisa do Hospital de Clínicas de Porto Alegre (FIPE-HCPA) to PA-P. The funders had no role in study design, data collection and analysis, decision to publish, or preparation of the manuscript.

**Competing interests:** The authors have declared that no competing interests exist.

suppressor gene encoding the p53 protein. Although any tumor can be identified in LFS carriers, "core" tumors of the syndrome have been reported and include premenopausal breast cancer, bone and soft-tissue sarcoma, brain cancer, leukemia and adrenocortical carcinoma (ACC). Carriers of germline *TP53* PV have a variable lifetime risk of developing cancer, and phenotype may vary from fully penetrant LFS to cancer-free over a lifetime. Nevertheless, about 50% of carriers develop at least one malignancy by age 30, especially those with *TP53* DNA-binding domain (DBD) variants, also called "classic" variants, which represent approximately 86% of the *TP53* pathogenic variants associated with the LFS phenotype in most countries [1–3].

Population prevalence studies have estimated that germline *TP53* PV occur at a frequency of 1 in 5,000 to 1 in 20,000 individuals [4]. In more recent studies, prevalence of *TP53* PV heterozygotes was proposed to reach 0.2% in Europeans [5, 6]. In addition, a germline *TP53* founder PV, c.1010G>A (p.Arg337His), widely referred as R337H, has been reported in Southern Brazil at a frequency of 1 in approximately 300 newborns [7–9], but tumor penetrance appears to be lower than that observed in carriers of DNA-binding domain (DBD) PV [10–13]. The arginine residue at codon 337 is involved in the protein oligomerization and functional data have shown that its replacement with histidine disrupts the tetramer form, making the domain unable to fully oligomerize in conditions of slightly elevated pH [14]. Although it was initially described as a "tissue-specific sequence variant" related only to ACC, today it is considered to be a PV related to the occurrence of multiple tumors, in a spectrum similar to that of LFS [15, 16]. Recent findings from a mouse model provided *in vivo* evidence that the R337H PV decreased p53 transactivation potential and renders mice susceptible to carcinogen-induced liver tumorigenesis [17].

Clinical criteria to define diagnosis of LFS were established based on the first study by Li and Fraumeni [18]. Approximately 70% to 80% of patients who fulfill classical criteria will have a germline PV in *TP53* [16, 19] When a broader LFS tumor spectrum was considered, a number of different sets of criteria started to be used to identify LFS patients, including the Chompret criteria and other criteria for Li-Fraumeni Like Syndrome (LFL) [19–21]. Importantly, diagnostic criteria defined by Chompret have increased the sensitivity of *TP53* germline PV detection by including patients with the core LFS tumors even without a family history. The revised Chompret criteria [21–23] had a PV detection rate of 18% and, when incorporated as part of *TP53* testing criteria along with classic LFS criteria, have been shown to improve the diagnostic sensitivity to 95% (Classic and Chompret criteria together) [2]. Therefore, the National Comprehensive Cancer Network (NCCN) and several other guidelines recommend using both the Classic LFS and the revised Chompret criteria to indicate germline *TP53* genetic testing [24].

So far, only a few studies showed the prevalence of germline *TP53* PV in individuals from Southern Brazil, in which the prevalence of 28,8% and 11,4% were found in a case series of 45 and 70 probands fulfilling any LFS criteria [25, 26]. In the present study, we aimed to characterize the clinical and molecular profile in a series of LFS patients fulfilling the 2015 revised Chompret criteria and recruited from cancer risk evaluation clinics in southern Brazil. These results can help to better define the LFS prevalence in Southern Brazil and also points out to differences in the clinical spectrum among carriers of distinct PV in *TP53*.

## Materials and methods

### Patients and ethical aspects

From July 2015 to January 2019, 211 independent cancer-affected patients from unique families with a suggestive clinical phenotype of LFS were identified at a public hospital and private cancer risk evaluation clinics in Southern Brazil. Of these, 191 were residents of the Southern region of Brazil, met the 2015 revised Chompret criteria and were included in the present

study. The majority of patients, 148 patients were from a reference public hospital (Hospital de Clínicas de Porto Alegre), seen at the institutional's outpatient cancer genetics clinic (108) and pediatric cancer ward (40). The additional 43 probands were identified in 4 private cancer genetics clinics in the same city. **S1 Fig** is a Consort Diagram that depicts the recruitment and testing process, while the **S1 Table** lists 2015 revised Chompret criteria. The study was approved by the Institutional Review Board. All participants underwent pre- and post-test genetic counseling, provided informed written or verbal consent for the study. When verbal consent was obtained, it was registered on participant clinical chart. Parents signed the consent for participants that were minors. Personal clinical history, self-reported family history and previous testing results were collected from patient interviews or medical records.

## Molecular analysis

Of the 191 patients participating in this study, 43 had previously undergone multi-gene panel testing (MGPT) including *TP53* sequence variant and rearrangement testing using Next-Generation Sequencing (NGS, retrospectively tested), 99 patients had undergone previous analysis of the *TP53* coding region by Sanger sequencing and Multiplex Ligation-Dependent Probe Amplification (MLPA) (also retrospectively tested), and 49 patients were offered molecular testing in the institutional research laboratory at recruitment (prospectively tested). *TP53* genotyping in the latter was performed employing two methodologies: (1) NGS in peripheral blood samples using the Ion AmpliSeq ™ Panel *TP53* kit (Thermo Fisher Scientific) and Ion GeneStudio S5 system (Ion Torrent Systems Inc, Gilford, NH); and (2) MLPA using the SALSA MLPA P056 kit (MRC Holland, Amsterdam, Netherlands), followed by capillary gel electrophoresis with the Applied Biosystem 3500 Genetic Analyzer (Applied Biosystems, Foster City, CA, USA) and analyses of the copy number variations conducted in the Coffalyser. Net software (MRC-Holland®) [27].

## Statistical analysis

Tumor spectrum and clinical characteristics of carriers of DBD variants (group A) and R337H variant (group B) were compared. Data normality assumptions were verified on the age of group A and B and Mann-Whitney-Wilcoxon test was performed. To measure the association among the groups, gender, type of cancer and multiple tumors we used Pearson's Chi-Squared test or Fisher's exact test. Odds ratio with 95% confidence intervals were also calculated. To compare the pathogenic variant detection rate in this study and the rate found in Bougeard et al in 2015 [2], we used Pearson's Chi-Squared test with Yates continuity correction. We also divided our probands in three groups (hotspot DBD variant carriers; R337H carriers and DBD non hotspot variant carriers) and Kruskal-Wallis test followed by Benjamini-Hochberg correction for multiple comparisons was performed. All data analyses were performed in R 3.4.2 statistical software.

## Results

Germline PV *TP53* were identified in twenty-six (13.6%) of the 191 probands included in the study. One of the carriers was homozygote and the other 25 carriers of germline PV were heterozygotes. 18 (69.2%) harboured the Brazilian founder R337H variant and 8 probands (30.8%) had a PV in the *TP53* DBD. MLPA analysis identified no *TP53* deletions and/or duplications in this series. **Fig 1** shows the location of each pathogenic alteration detected in the gene and **Table 1** summarizes the clinical and molecular results of all PV-positive probands (**S2 Table** exhibits the characterization of all probands analyzed). **Fig 2** depicts the NGS results encompassing the entire *TP53* coding region from two probands.

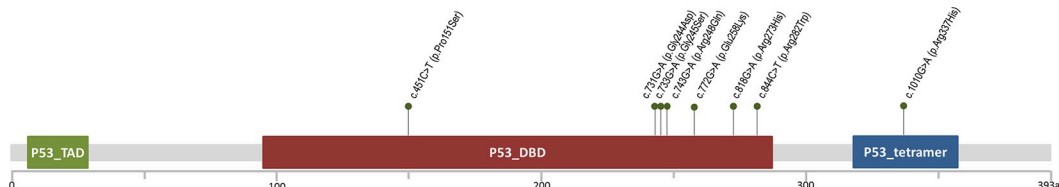

**Fig 1. Location of the *TP53* pathogenic variants detected in the *p53* protein functional domains.** Green dots represent the variants identified in the present study. P53_TAD, transactivation domain; P53_DBD, DNA binding domain; P53_oligomer, oligomerization domain.

Important differences were observed when comparing the tumor spectrum and clinical characteristics of carriers of DBD variants (group A) and R337H variant (group B) (**Table 2**). The median age at first cancer diagnosis was 22 years in group A and 2 years in group B (*P* = 0.009; Mann-Whitney-Wilcoxon test). Fifteen patients (83.3%) in group B and only 3 (37.5%) in group A developed a tumor before age 18 years. Most of the tumors (13, 72.22%) observed in group B were ACC (all under 18 years), and only one ACC (12.5%) was observed in group A (diagnosed at age 44 years). Finally, multiple primary tumors were observed only among patients from group A, including 4 (50%) patients. Interestingly, one proband had been diagnosed with 4 primary tumors: osteosarcoma, bilateral breast cancer and soft tissue sarcoma; all before age 25 years. The tumor spectrum of the PV carriers is depicted in **Fig 3** and shows evident differences between groups (DBD variant and R337H carriers), especially regarding ACC.

As observed in **Table 3**, a significant association was found in the comparative analyses including type of cancer and multiple tumors. A higher prevalence of ACC was observed in group B when compared to group A patients (*P* = 0.043; Chi-squared test) and the presence of multiple tumors was most frequent in group A (*P* = 0.005, Fisher exact test). Additionally, we classified the DBD variants in two groups, namely: group A non-hotspot PV, which comprised of p.(Pro151Ser), p.(Gly244Asp) and p.(Glu258Lis) variants; and group A hotspot PV (p.(Gly245Ser), p.(Arg248Gln), p.(Arg273His), p.(Arg282Trp)). When comparing the median age at first diagnosis of cancer in patients from group A non-hotspot PV, group A hotspot PV, and group B (R337H variants) we observed a significative difference (*P* = 0.021), being 31.8, 12.1 and 2.35 years respectively. The post-hoc analysis pointed out that age at first diagnosis was different between group B and A non-hotspot PV (data not shown).

## Discussion

LFS is considered a rare cancer predisposition disorder worldwide. In Southern Brazil, due to presence of a germline founder pathogenic variant in the *TP53* oligomerization domain (R337H), it is estimated that 0,3% of the general population carries this variant [12]. Despite significant heterozygote frequency at the population level, little information is available on the prevalence of germline *TP53* PV among individuals with a suggestive phenotype, i.e. fulfilling revised Chompret criteria. This information is important to guide health care policies for cancer prevention and treatment in the region. Identifying LFS patients is important to determine adequate clinical surveillance and follow up, not only in the proband but in his/her relatives, since detection of a carrier provides the opportunity for cascade testing and, if additional carriers are identified in the family, they can be referred to appropriate genetic counseling and specific high risk screening protocols [28]. Villani and colleagues (2016) demonstrated that carriers of pathogenic *TP53* variants benefit enormously from an enhanced surveillance protocol, including frequent physical examination plus targeted biochemical monitoring and

**Table 1. Clinical and molecular characterization of all LFS probands harboring germline *TP53* pathogenic variants (PV) identified in this study.**

| Proband ID / Gender | Age at 1st cancer diagnosis (years) | Proband's type of cancer | Age at diagnosis, other tumors (years) | 2015 Version Chompret Criterion(s) | Recruitment | Genetic Testing | chr17 position on Assembly GRCh37 (dbSNP rs ID) | *TP53* variant HGVS c. | *TP53* variant HGVS p. |
|---|---|---|---|---|---|---|---|---|---|
| 166 / F | 32 | Breast | Breast (38) | Familial | PC | Sanger + MLPA | rs28934874 | c. 451C>T | p. (Pro151Ser) |
| 167 / F | 30 | Breast (bilateral) | Thyroid (37) | Familial, EOBC | PUB | Sanger + MLPA | rs1057517983 | c.731G>A | p. (Gly244Asp) |
| 168 / F | 11 | CNS | NA | Familial | PUB | Sanger + MLPA | rs28934575 | c.733G>A | p. (Gly245Ser) |
| 169 / F | 12 | OS | Breast (21), Breast (22), STS (24) | MT, EOBC | PC | MGPT | rs28934575 | c.733G>A | p. (Gly245Ser) |
| 170 / F | 25 | Breast | NA | EOBC | PC | MGPT | rs11540652 | c.743G>A | p. (Arg248Gln) |
| 171 / M | 44 | ACC | NA | RT | PUB | NGS + MLPA | rs121912652 | c.772G>A | p. (Glu258Lys) |
| 172 / F | 19 | OS | Breast (29), STS (38) | Familial, MT, EOBT | PUB | Sanger + MLPA | rs28934576 | c.818G>A | p. (Arg273His) |
| 173 / F | 5 | CNS (CPC) | NA | RT | PC | MGPT | rs28934574 | c.844C>T | p. (Arg282Trp) |
| 174 / F | 0 (6 mo) | ACC | NA | RT | PED | Sanger + MLPA | rs121912664 | c.1010G>A | p. (Arg337His) |
| 175 / F | 0 (4 mo) | ACC | NA | Familial, RT | PED | Sanger + MLPA | rs121912664 | c.1010G>A | p. (Arg337His) |
| 176 / F | 0 (8 mo) | ACC | NA | RT | PED | Sanger + MLPA | rs121912664 | c.1010G>A | p. (Arg337His) |
| 177 / F | 1 | ACC | NA | Familial, RT | PUB | Sanger + MLPA | rs121912664 | c.1010G>A | p. (Arg337His) |
| 178 / M | 1 | ACC | NA | RT | PED | Sanger + MLPA | rs121912664 | c.1010G>A | p. (Arg337His) |
| 179 / M | 2 | ACC | NA | RT | PED | Sanger + MLPA | rs121912664 | c.1010G>A | p. (Arg337His) |
| 180 / M | 2 | ACC | NA | RT | PUB | Sanger + MLPA | rs121912664 | c.1010G>A | p. (Arg337His) |
| 181 / F | 3 | ACC | NA | RT | PUB | Sanger + MLPA | rs121912664 | c.1010G>A | p. (Arg337His) |
| 182 / F | 3 | ACC | NA | RT | PUB | NGS + MLPA | rs121912664 | c.1010G>A | p. (Arg337His) |
| 183 / F | 5 | ACC | NA | RT | PED | Sanger + MLPA | rs121912664 | c.1010G>A | p. (Arg337His) |
| 184 / F | 11 | ACC | NA | RT | PED | Sanger + MLPA | rs121912664 | c.1010G>A | p. (Arg337His) |
| 185 / M | 17 | ACC | NA | RT | PED | Sanger + MLPA | rs121912664 | c.1010G>A | p. (Arg337His) |
| 186 / F | 23 | Breast | NA | Familial, EOBC | PUB | Sanger + MLPA | rs121912664 | c.1010G>A | p. (Arg337His) |
| 187 / F | 57 | Breast | NA | Familial | PUB | Sanger + MLPA | rs121912664 | c.1010G>A | p. (Arg337His) |
| 188 / F | 49 | Breast (bilateral) | NA | Familial | PUB | Sanger + MLPA | rs121912664 | c.1010G>A | p. (Arg337His) |
| 189 / M | 1 | CNS (CPC) | NA | RT | PED | Sanger + MLPA | rs121912664 | c.1010G>A | p. (Arg337His) |
| 190 / M | 1 | CNS (CPC) | NA | RT | PED | Sanger + MLPA | rs121912664 | c.1010G>A | p. (Arg337His) |

(*Continued*)

**Table 1.** (*Continued*)

| Proband ID / Gender | Age at 1st cancer diagnosis (years) | Proband's type of cancer | Age at diagnosis, other tumors (years) | 2015 Version Chompret Criterion(s) | Recruitment | Genetic Testing | chr17 position on Assembly GRCh37 (dbSNP rs ID) | *TP53* variant HGVS c. | *TP53* variant HGVS p. |
|---|---|---|---|---|---|---|---|---|---|
| 191 / F | 1 | ACC | NA | Familial, RT | PED | Sanger + MLPA | rs121912664 | c.1010G>A | p. (Arg337His)* |

ACC, Adrenocortical Carcinoma; CNS, Central Nervous System; CPC, Choroid Plexus Carcinoma; EOBC, Early Onset Breast Cancer; MGPT, Multigene Panel Testing; MT, Multiple Tumors; MO, months old; OS, Osteosarcoma; RT, Rare Tumors, STS, Soft tissue sarcoma; NA, not applicable; PUB, high-risk public clinic; PC, high-risk private clinic; PED, pediatric tumors database; NGS, Next-generation Sequencing; MLPA, Multiplex Ligation-Dependent Probe Amplification; WT, wild-type genotype
* homozygous for the R337H variant.

periodic imaging screens (ultrasounds, brain magnetic resonance images, and rapid whole body MRI scans) [28]. Collectively, this approach has a significant impact in overall survival, compared to patients that do not undergo enhanced surveillance. In Brazil, although patients with health insurance have access to genetic testing if they fulfill the revised Chompret criteria, those that rely solely on the public health care system (about 70% of the population) must pay out of pocket to have this information, since genetic testing for cancer predisposition is not yet payed in the public setting.

In this cohort, tumoral spectrum in R337H carriers was similar to that already described in literature, especially when compared to previous studies performed in other Brazilian Centers. However, in the present study a strikingly higher prevalence of ACC was observed in R337H carriers when compared to carriers of DBD variants ($P = 0.043$; chi-squared test). From this observation we can conclude that in the series presented here, ACC was the most prevalent tumor observed in association with R337H whereas the previous Brazilian study described breast cancer as the most frequent tumor (30%) [25].

Regarding PV detection rate for the 2015 Chompret Criteria identified here (13,6%), this rate is similar to the 18% described by Bougeard *et al.* in 2015 in France ($P = 0.2482$; chi-squared test with Yates correction), but it is mainly due to the presence of the R337H variant [2]. Of note, in the previous study by Andrade *et al.* (2017) of Brazilian patients from the Southeastern region, PV detection rate in 17 probands with the 2015 Chompret Criteria was much higher, 35% [26]. These differences between the studies from Southern and Southeastern Brazil may reflect regional genetic modifiers of the phenotype (i.e. additional genetic risk factors), regional environmental factors or different recruitment strategies in each study.

Regarding genotype-phenotype correlations, it is well know that DBD hotspot variants with reported dominant negative effects, such as p.(Gly245Ser), p.(Arg248Gln), p.(Arg273His) and p.(Arg282Trp) are associated with earlier onset cancers and stronger family history of tumors within the LFS spectrum [29]. On the other hand, several previous studies from Brazilian cohorts have suggested that R337H is a PV with lower prevalence associated with cancer diagnoses at older ages, although a bimodal distribution of age at cancer diagnosis has also been suggested [30, 31]. Contrary to the expected phenotype, probands with the R337H variant in this study had earlier age at first tumor diagnosis when compared with carriers of DBD variants. To analyze this data in more detail, we divided our probands in three groups according to type of PV (non-hotspot DBD, hotspot DBD and R337H) and observed that median age at first tumor diagnosis among groups with the lowest mean age identified among R337H carriers.

The results of the present study are relevant for two main reasons. First, they underscore the importance of considering that significant regional differences may occur and that criteria

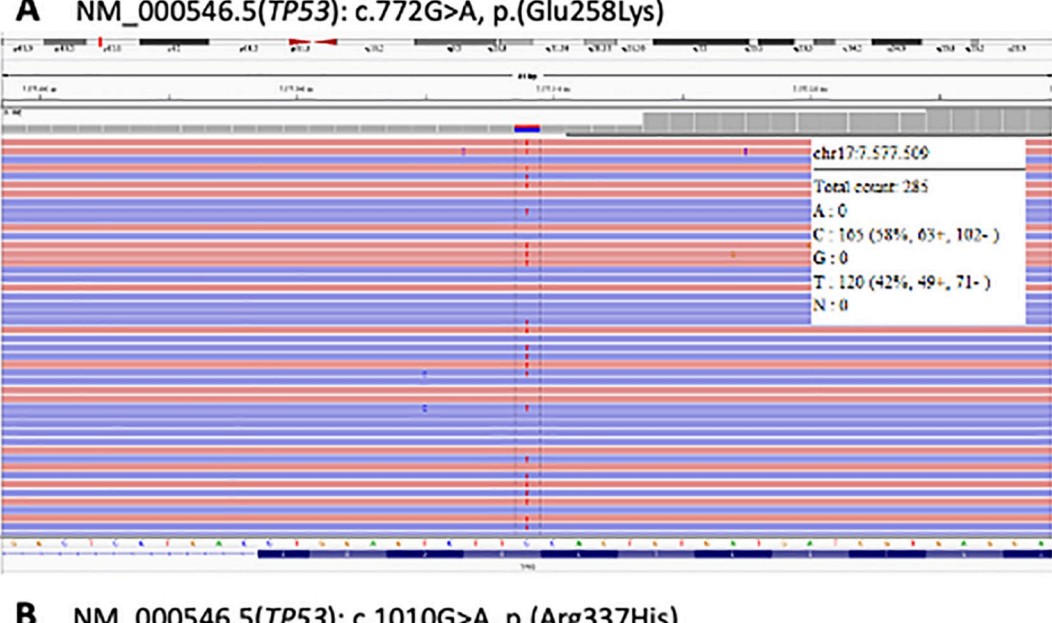

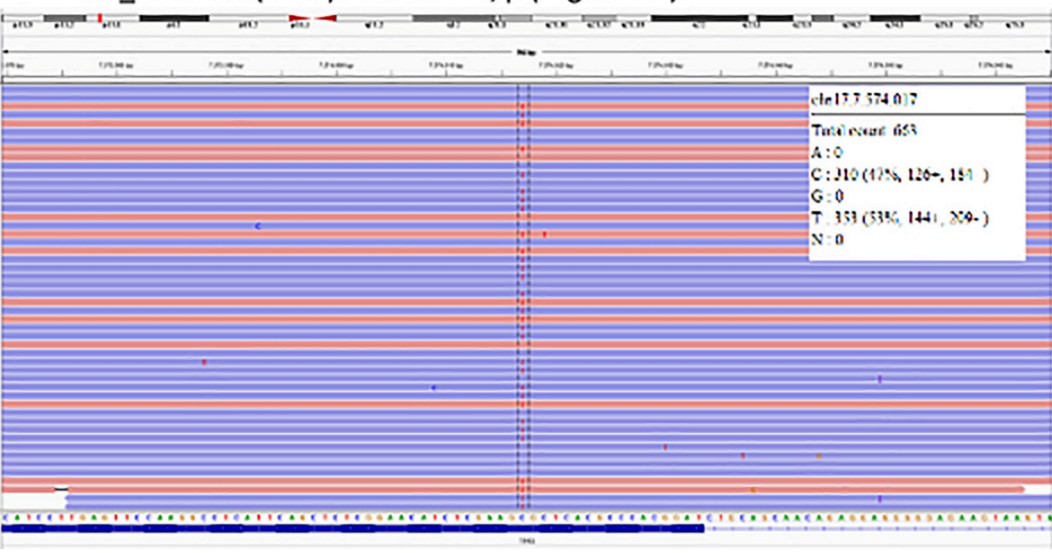

**Fig 2. Representative next-generation sequencing results encompassing the *TP53* entire coding region (minimum coverage of 100X by amplicon) from two probands fulfilling the 2015 revised Chompret criteria for Li-Fraumeni syndrome.** (A) Carrier of a germline pathogenic variant (PV) located in the p53 DNA binding domain (DBD); and (B) carrier of the Brazilian founder R337H PV located in the p53 oligomerization domain. Description of *TP53* sequence variants is provided according to updated Human Genome Variation Society (HGVS) recommendations. Human *TP53* sequence corresponding to the NM_000546.5 was used as a wild-type reference. Right panels show wild-type and variant allele counts, which were consistent with the expected germline occurrence of these genetic alterations (around 50% of reads for each allele). Note that both alleles were analyzed from antisense strand due to the *TP53* gene orientation. Chr17, position or genomic coordinate at chromosome 17 (GRCh37/hg19 human genome assembly).

established for one population may not have the same performance in another population. Considering that the population of Southern Brazil is mostly of European ancestry, one would expect to see a prevalence of germline *TP53* PV variants similar to that observed in Europeans. A high frequency of R337H among probands with a phenotype suggesting LFS had been previously reported by Achatz et al. (2007) (46,1% of those with coding region *TP53* variants), but

**Table 2. Distribution of tumor types in all LFS PV-positive patients (n = 26).**

| Tumor types diagnosed in PV carriers | Number of tumors per PV group (A/B) | OR (95% CI), p value | Number of patients per group (A/B) | % PV carriers per group (A/B) | Age at diagnosis (range when >1) in each group (A/B) |
|---|---|---|---|---|---|
| Adrenocortical Carcinoma | 1 / 13 | 16.0 (1.5–875.8), **0.009** | 1 / 13 | 12.5 / 72.2 | 44 / 0 to 17 |
| Breast | 8 / 4 | 0.18 (0.03–1.0), **0.03** | 5 / 3 | 62.5 / 16.6 | 21 to 38 / 23 to 57 |
| CNS | 2 / 2 | 0.39 (0.02–6.53), 0.56 | 2/ 2 | 25 / 11.1 | 5 to 11 / 1 |
| Osteosarcoma | 3 / NA | - | 2 / NA | 25 / NI | 12 to 38 / NA |
| Thyroid | 1 / NA | - | 1 / NA | 12.5 / NI | 37 / NA |

DBD, pathogenic variants located in the DNA-binding domain; CNS, Central Nervous System tumors; NA, not applicable; NI, not identified.

these authors did not restrict their recruitment to patients fulfilling Chompret criteria [25]. Second, results from the present analysis, in which overall *TP53* germline PV detection rate in Chompret criteria fulfilling probands was lower than expected from previous studies, may suggest that a different set of pathogenic variants, not yet mapped (i.e. located in intronic or regulatory regions of *TP53*) may be associated with the LFS phenotype in this particular region. It is also possible that PV in other, yet unidentified genes are associated with the LFS phenotype, accounting for the "missing heritability" of more than 85% observed here [32, 33]. An important limitation of the present study, that must be accounted for when analyzing the results is this study, is that a significant proportion of data on genetic testing were obtained retrospectively and with different variant detection strategies. Thus, further analyses on a prospectively recruited cohort of probands fulfilling Chompret criteria and then, clinical assessment of families carrying either DBD PV or R337H will be important to confirm these findings. Expanding

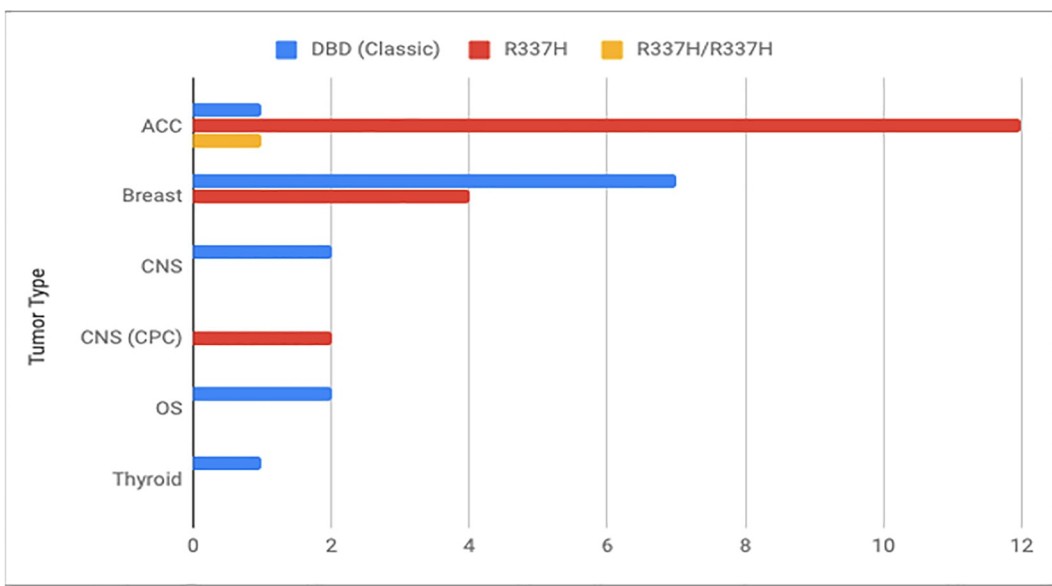

**Fig 3. Graphic showing the differences between the tumor spectrum observed in carriers of the DBD variants, R337H variant and R337H homozygous proband.** ACC, adrenocortical carcinoma; CNS, central nervous system; CPC, choroid plexus carcinoma; DBD, DNA binding domain; OS, osteosarcoma.

**Table 3. Association of gender, age at first tumor diagnosis, tumor type and development of multiple tumors among carriers of different groups of germline PV *TP53* (groups A and B).**

| | Group of pathogenic germline variants (PV) | | |
|---|---|---|---|
| | A (n = 8) | B (n = 18) | *P* value |
| **Gender** | | | |
| Female | 7 | 12 | 0.375* |
| Male | 1 | 6 | |
| **Age at first cancer diagnosis, median (IQR)** | 22 (11.7–30.5) | 2.0 (1.0–9.5) | **0.009**** |
| **Tumor types** | | | |
| ACC | 1 | 13 | **0.043†** |
| Breast | 2 | 2 | |
| Breast bilateral | 1 | 1 | |
| CNS | 1 | 0 | |
| CNS (CPC) | 1 | 2 | |
| OS | 2 | 0 | |
| **Multiple tumors** | | | |
| Yes | 4 | 0 | **0.005*** |
| No | 4 | 18 | |

† Pearson Chi-squared test.

* Fisher exact test.

** Mann-Whitney-Wilcoxon test.

ACC, adrenocortical carcinoma; OS, osteosarcoma; CNS, central nervous system; CPC, choroid plexus carcinoma.

this study in the region will be essential to instrument policy makers in establishing cancer screening protocols for these individuals.

## Conclusions

The current study shows the impressive clinical heterogeneity of LFS, highlights particularities of the founder *TP53* pathogenic variant R337H and points to the need for larger and collaborative studies to better define LFS prevalence, clinical spectrum and penetrance of different types of PVs in the Brazilian population.

## Supporting information

**S1 Fig. Consort Diagram representing the patient recruitment and genetic testing process employed in the current study.**
(DOCX)

**S1 Table. 2015 revised Chompret criteria for LFS and *TP53* gene testing.**
(DOCX)

**S2 Table. Clinical and molecular characterization of all probands (n = 191) included in the study.**
(DOCX)

## Acknowledgments

We would like to thank Gustavo Stumpf da Silva and Patricia Santos-Silva for their valuable contributions and technical support.

## Author Contributions

**Conceptualization:** Camila Matzenbacher Bittar, Igor Araujo Vieira, Cristina B. O. Netto, Barbara Alemar, Gabriel S. Macedo, Patricia Ashton-Prolla.

**Data curation:** Yasminne Marinho de Araújo Rocha, Clévia Rosset, Cristina B. O. Netto, Gabriel S. Macedo, Patricia Ashton-Prolla.

**Formal analysis:** Camila Matzenbacher Bittar, Igor Araujo Vieira, Clévia Rosset, Tiago Finger Andreis, Patricia Ashton-Prolla.

**Funding acquisition:** Patricia Ashton-Prolla.

**Investigation:** Camila Matzenbacher Bittar, Yasminne Marinho de Araújo Rocha, Igor Araujo Vieira, Clévia Rosset, Tiago Finger Andreis, Ivaine Tais Sauthier Sartor, Osvaldo Artigalás, Cristina B. O. Netto, Barbara Alemar, Gabriel S. Macedo, Patricia Ashton-Prolla.

**Methodology:** Camila Matzenbacher Bittar, Yasminne Marinho de Araújo Rocha, Igor Araujo Vieira, Clévia Rosset, Tiago Finger Andreis, Ivaine Tais Sauthier Sartor, Barbara Alemar, Gabriel S. Macedo, Patricia Ashton-Prolla.

**Project administration:** Camila Matzenbacher Bittar.

**Resources:** Osvaldo Artigalás, Cristina B. O. Netto.

**Software:** Yasminne Marinho de Araújo Rocha, Ivaine Tais Sauthier Sartor.

**Supervision:** Patricia Ashton-Prolla.

**Validation:** Yasminne Marinho de Araújo Rocha.

**Writing – original draft:** Camila Matzenbacher Bittar, Patricia Ashton-Prolla.

**Writing – review & editing:** Camila Matzenbacher Bittar, Igor Araujo Vieira, Clévia Rosset, Tiago Finger Andreis, Osvaldo Artigalás, Cristina B. O. Netto, Barbara Alemar, Gabriel S. Macedo, Patricia Ashton-Prolla.

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
