## [Decision Letter · Decision Letter 0]

5 Jan 2021

PONE-D-20-38705

Clinical and molecular characterization of patients fulfilling Chompret criteria for Li-Fraumeni Syndrome in Southern Brazil

PLOS ONE

Dear Dr. Ashton-Prolla,

Thank you for submitting your manuscript to PLOS ONE. After careful consideration, we feel that it has merit but does not fully meet PLOS ONE’s publication criteria as it currently stands. Therefore, we invite you to submit a revised version of the manuscript that addresses the points raised during the review process.

1.  Include a CONSORT diagram to describe participant inclusion and exclusion.  Provide clarity on whether all participants were from unique families or if there were any related cases.  See additional comments from reviewers on descriptions of inclusion/exclusion.

2. Consider adding in a new table to describe the overall cohort of the 191 study participants in a more concise manner with percentages etc. See both reviewer's comments.

3. Replace mutation with pathogenic variant per recently revised genetics terminology.

4.  Provide more description on the technology used for pathogenic variant detection.  If multiple methods were used were there differences in types of variants detected or rate of variant detection?

5.  Update the Discussion per the reviewer's suggestions.

6. Consider adding in the suggested references.

7.  Address both reviewer's comments related to putting this study into the context of the literature.

We look forward to receiving your revised manuscript.

Kind regards,

Amanda Ewart Toland, Ph.D.

Academic Editor

PLOS ONE

Journal Requirements:

2. Please provide additional details regarding participant consent. In the ethics statement in the Methods and online submission information, please ensure that you have specified:

 - whether consent was informed

 - when oral consent was obtained, how was it documented and witnessed

 - whether parents orr guardians signed the consent form for minors

3. Please provide the names of the cancer evaluation clinics.

Reviewers' comments:

Reviewer's Responses to Questions

**Comments to the Author**

1. Is the manuscript technically sound, and do the data support the conclusions?

Reviewer #1: Yes

Reviewer #2: Yes

2. Has the statistical analysis been performed appropriately and rigorously? 

Reviewer #1: I Don't Know

Reviewer #2: Yes

3. Have the authors made all data underlying the findings in their manuscript fully available?

Reviewer #1: No

Reviewer #2: Yes

4. Is the manuscript presented in an intelligible fashion and written in standard English?

Reviewer #1: Yes

Reviewer #2: Yes

5. Review Comments to the Author

Reviewer #1: This is an interesting manuscript which continues to add to our dynamic understanding of germline TP53 PVs. The manuscript would be aided by the following additions/changes:

-Please replace use of the term mutation in all places as this is no longer the preferred terminology.

-There are some notable omissions from the reference list that would be best cited including the following by LFS experts (David Malkin/ Judy Garber):

—>In particular during the discussion of the oligomerization domain: https://www.ncbi.nlm.nih.gov/pmc/articles/PMC6292786/

Lines 68-74: Reference work by Rana et al.

https://www.ncbi.nlm.nih.gov/pmc/articles/PMC6292786/

https://pubmed.ncbi.nlm.nih.gov/31105275/

-On variable penetrance of TP53 and meeting of Chompret criteria: You can contrast your findings in the discussion as there are notable differences likely based in the ascertainment of the cohorts studied, but it is worth contrasting these results.

-Why were some patients excluded? This is not explained. Please include a CONSORT diagram on how the 191 were arrived at.

-Please limit Table 1 to the TP53 pathogenic and likely pathogenic (PV) carriers only. The negative ones can be in a supplemental table.

Table 2: Instead of referring to mutations as ‘classic’ can you label them as DBD. Use of ‘classic’ in this context and in the table is confusing.

Relative risks or Odds ratios with 95% confidence intervals should be provided.

Table 3: For cohorts a/b: include median age at testing with IQR; include median age at first cancer diagnosis also with IQR.

Line 214 — comparison to literature should be in the discussion and not in the results section

Line 243 —include discussion of ascertainment

Line 250 — poor prognosis for what? Overall survival?

My understanding is that R337H was first identified due to the unusually high frequency of ACC, there is no mention of this well-established association or prior literature supporting the role of this variant in ACC. For example, https://pubmed.ncbi.nlm.nih.gov/15952083/ from 2005.

The discussion would be aided by referencing the findings of Pinto et al., which may help to explain potential modifiers of TP53 R337H : https://pubmed.ncbi.nlm.nih.gov/32637605/ from 2020.

Reviewer #2: A good study examining the frequency of TP53 pathogenic variants (PV) in patients with cancer meeting the revised 2015 Chrompret criteria in Southern Brazil. The investigation of 191 cases builds the case for differences between age of onset and prevalence of cancer type between group A (classic mutations in the DNA binding domain of TP53) and group B (oligomerization domain encompassing the prevalent R337H PV in Brazil). Findings show a PV detection rate of 13.6% and that ACC is more prevalent in group B with a younger age of onset then other previous studies have suggested.

Comment one:

The abstract mentions “191 cancer affected and unrelated probands” (line 36) does unrelated probands mean all index cancer cases were independent and not from the same family? If so consider re-wording to “191 independent cancer cases from unique families”. The introduction would be aided with inclusion of literature on the prevalence of TP53 cases identified from the expansion of multi-gene panel testing (MGPT) in clinical practice and how this relates to targeting TP53 testing to cases that meet classic TP53 testing criteria and broadening of testing criteria.

Comment two:

In the methods section, it is not clear whether MGPT, Sanger sequencing or research testing using next generation sequencing (NGS) was applied retrospectively or just data collected retrospectively on a consecutive case series. This point needs clarifying to understand the origins of the data. If from retrospective data or application of testing retrospectively this needs to be mentioned in the limitations of the study also.

The methods section should include a statement in the first paragraph that all cancer cases and family histories meet revised 2015 Chrompret criteria and were selected for on this basis.

Comment three:

The results section would be aided with a concise summary description of the population characteristics. Consider the expansive Table 1 as a supplementary table and instead summarise this data in a concise table with percentages for cancer type, gender, average age or age range of diagnosis, which aspect of chrompret criteria was meet, testing strategy and result. Suggest to organise case characteristics by negative and positive result or by group A or B. In this re-organisation it would be useful to understand the percentage of TP53 cases picked up by testing regimes MGPT, sanger or research NGS. This would link back to the inclusion of literature in introduction of TP53 pick up from MGPT in general. The existing supplementary table 1 defining the revised 2015 Chrompret criteria could be used alongside the above data to re-organise or link to expansive table 1 when moved to supplementary material. This would help the reader to digest and interpret the results better.

Comment four:

For the reader the discussion would be aided by stating the main findings of the study from the start and then discuss each main finding in separate paragraphs. Inclusion of the emerging screening modalities for TP53 carriers ie MRI screening in adult TP53 carriers and paediatric screening protocol with attention to efficacy and benefit in the context of Brazil’s current funding model of genetic testing access and screening limits. There isn’t a comprehensive limitations section – need to include additional information on limitations ie study design in the nature of methods and statistical tools used and link to the future research direction information ie the use of different study designs in the application of MGPT with population control and cases included and other avenues of investigation to confirm the higher prevalence of R337H in group B and ACC with different ethnicities and larger data sets.

6. PLOS authors have the option to publish the peer review history of their article (what does this mean?). If published, this will include your full peer review and any attached files.

Reviewer #1: No

Reviewer #2: No

---

## [Author Response · Author response to Decision Letter 0]

27 Mar 2021

We have attached a letter containing all answers to the reviewer's and editor's comments

---

## [Decision Letter · Decision Letter 1]

30 Apr 2021

Clinical and molecular characterization of patients fulfilling Chompret criteria for Li-Fraumeni Syndrome in Southern Brazil

PONE-D-20-38705R1

Dear Dr. Ashton-Prolla,

We’re pleased to inform you that your manuscript has been judged scientifically suitable for publication and will be formally accepted for publication once it meets all outstanding technical requirements.

Kind regards,

Amanda Ewart Toland, Ph.D.

Academic Editor

PLOS ONE

Additional Editor Comments (optional):

Reviewers' comments:

Reviewer's Responses to Questions

**Comments to the Author**

1. If the authors have adequately addressed your comments raised in a previous round of review and you feel that this manuscript is now acceptable for publication, you may indicate that here to bypass the “Comments to the Author” section, enter your conflict of interest statement in the “Confidential to Editor” section, and submit your "Accept" recommendation.

Reviewer #2: All comments have been addressed

2. Is the manuscript technically sound, and do the data support the conclusions?

Reviewer #2: Yes

3. Has the statistical analysis been performed appropriately and rigorously? 

Reviewer #2: Yes

4. Have the authors made all data underlying the findings in their manuscript fully available?

Reviewer #2: Yes

5. Is the manuscript presented in an intelligible fashion and written in standard English?

Reviewer #2: Yes

6. Review Comments to the Author

Reviewer #2: The author has responded to comments - Table 1 presents in it's original form in the main manuscript as per the first submission and is now also in supplementary. Consider, removing from main manuscript and refer to supplementary table 2 in the text or keep table and revise to only include the description of cases that were identified with PV as these are further described in the main manuscript.

7. PLOS authors have the option to publish the peer review history of their article (what does this mean?). If published, this will include your full peer review and any attached files.

Reviewer #2: No

---

## [Editor Report · Acceptance letter]

25 Aug 2021

PONE-D-20-38705R1 

Clinical and molecular characterization of patients fulfilling Chompret criteria for Li-Fraumeni Syndrome in Southern Brazil 

Dear Dr. Ashton-Prolla:

I'm pleased to inform you that your manuscript has been deemed suitable for publication in PLOS ONE. Congratulations! Your manuscript is now with our production department. 

Kind regards, 

on behalf of

Dr. Amanda Ewart Toland 

Academic Editor

PLOS ONE